

# Genome-wide characterization of the auxin response factor (ARF) gene family of litchi (*Litchi chinensis* Sonn.): phylogenetic analysis, miRNA regulation and expression changes during fruit abscission

Yanqing Zhang, Zaohai Zeng, Chengjie Chen, Caiqin Li, Rui Xia and Jianguo Li

State Key Laboratory for Conservation and Utilization of Subtropical Agro-Bioresources, Guangdong Litchi Engineering Research Center, South China Agricultural University, Guangzhou, Guangdong, China

## ABSTRACT

Auxin response factors (ARFs) play fundamental roles in modulating various biological processes including fruit development and abscission via regulating the expression of auxin response genes. Currently, little is known about roles of ARFs in litchi (*Litchi chinensis* Sonn.), an economically important subtropical fruit tree whose production is suffering from fruit abscission. In this study, a genome-wide analysis of ARFs was conducted for litchi, 39 ARF genes (*LcARFs*) were identified. Conserved domain analysis showed that all the LcARFs identified have the signature B3 DNA-binding (B3) and ARF (Aux_rep) domains, with only 23 members having the dimerization domain (Aux_IAA). The number of exons in LcARF genes ranges from 2 to 16, suggesting a large variation for the gene structure of *LcARFs*. Phylogenetic analysis showed that the 39 LcARFs could be divided into three main groups: class I, II, and III. In total, 23 *LcARFs* were found to be potential targets of small RNAs, with three conserved and one novel miRNA-*ARF* (miRN43-*ARF9*) regulatory pathways discovered in litchi. Expression patterns were used to evaluate candidate *LcARFs* involved in various developmental processes, especially in flower formation and organ abscission. The results revealed that most ARF genes likely acted as repressors in litchi fruit abscission, that is, *ARF2D/2E, 7A/7B, 9A/9B, 16A/16B*, while a few *LcARFs*, such as *LcARF5A/B*, might be positively involved in this process. These findings provide useful information and resources for further studies on the roles of ARF genes in litchi growth and development, especially in the process of fruit abscission.

## INTRODUCTION

Auxin plays a central role in numerous aspects of plant developmental and physiological processes, including embryogenesis, apical dominance, vascular elongation, flowering, fruit development, and lateral root initiation (*Woodward & Bartel, 2005*; *Fleming, 2006*). Auxin response factors (ARFs) are a group of important transcription factors in the

Corresponding authors
Rui Xia, rxia@scau.edu.cn
Jianguo Li, jianli@scau.edu.cn

auxin signaling pathway, which can activate or repress the expression of early/primary auxin response genes by binding to the auxin response element (AuxRE) site in their promoter regions (*Liscum & Reed, 2002*; *Guilfoyle & Hagen, 2007*). A typical ARF is characterized by a highly conserved N-terminal B3-type DNA binding domain (DBD) that recognized the AuxRE motif, an activation domain or repression domain, and a carboxy-terminal dimerization domain (domain III/IV), which is involved in protein–protein interactions by dimerizing with auxin/indole-3-acetic acid (Aux/IAA) family genes (*Kim, Harter & Theologis, 1997*; *Guilfoyle & Hagen, 2007*; *Piya et al., 2014*).

Auxin response factors exert pivotal function in the regulation of plant growth and development through the auxin signaling pathway (*Kepinski & Leyser, 2005*; *Li et al., 2016*). Due to the significance of ARFs, genome-wide characterization of ARFs have been completed in many species such as model plants *Arabidopsis* (*Hagen & Guilfoyle, 2002*) and *Solanum lycopersicum* (*Zouine et al., 2014*), important crops such as soybean (*Ha et al., 2013*; *Le et al., 2016*) maize (*Xing et al., 2011*; *Wang et al., 2012*), fruit trees like citrus (*Citrus sinensis*) (*Xie et al., 2015*) apple (*Malus domestica*) (*Luo et al., 2014*), and so on. From embryogenesis to flowering, mutants in members of ARFs exhibit diverse phenotypes, which show their unique and redundant functions for plant development. For instance, in *Arabidopsis thaliana*, *arf1* and *arf2* mutations affect leaf senescence and floral organ abscission (*Ellis et al., 2005*) and the loss of *AtARF3* causes defects in gynoecium patterning (*Nemhauser, Feldman & Zambryski, 2000*; *Liu et al., 2014a*). AtARF5 influences embryo, root, and shoot development (*Krogan et al., 2012*; *Crawford et al., 2015*). AtARF9 acts in suspensor cells to mediate hypophysis specification (*Rademacher et al., 2012*), and AtARF10/16/17 play vital roles in negatively regulating seed germination and post-germination activities (*Liu et al., 2007*).

Recently, small RNAs, especially miRNAs, have been emerging as critical regulators in almost all aspects of plant growth and development. Many members of the ARF family have been reported to be targets of miRNAs. In *Arabidopsis*, *AtARF6* and *AtARF8* are targets of miR167 (*Wu, Tian & Reed, 2006*), *AtARF10/16/17* are targets of miR160 (*Liu et al., 2007*, *2010*), and *ARF2/3/4* are targets of trans-acting siRNAs (tasiRNAs) generated from miR390-targeted TAS3 gene (trans-acting siRNA gene 3) (*Allen et al., 2005*; *Axtell et al., 2006*). These targeting relationships of miRNA on ARF genes are widely conserved in land plants (*Xia, Xu & Meyers, 2017*) and also in horticultural plants (*Chen et al., 2018b*). It has been reported that down-regulation of *ARF6* and *ARF8* by miR167 leads to floral development defects and female sterility in tomatoes (*Liu et al., 2014b*). Down-regulation of sly-miR160, increasing the expression of its targets *SlARF10/16/17*, regulates auxin-mediated ovary patterning as well as floral organ abscission and lateral organ lamina outgrowth (*Damodharan, Zhao & Arazi, 2016*). In *Arabidopsis* and tomato, the tasiRNA-mediated regulation of *ARF2* is involved in controlling the onset of leaf senescence and floral organ abscission (*Ellis et al., 2005*; *Lim et al., 2010*; *Guan et al., 2014*; *Ren et al., 2017*).

Litchi (*Litchi chinensis* sonn.), an important economic fruit tree in southern China, usually undergo serious fruit abscission before harvest, leading to a low yield. Many studies demonstrate *ARFs* play critical roles in regulating plant organ abscission (*Ellis et al., 2005*;

*Kuang et al., 2012*; *Guan et al., 2014*; *Xie et al., 2015*; *Xu et al., 2015*), while which *ARFs* are involved or more important than other ARFs in the fruit abscission in litchi remains elusive. Here, we identified 39 ARF genes in litchi. Gene structure, phylogeny, and targeting relationship with miRNAs were characterized. The expression of *LcARFs* was examined in diverse tissues and in the process of fruit abscission which was induced by three different treatments. Among them, *ARF2D/2E*, *7A/7B*, *9A/9B*, *16A/16B*, and *LcARF5A/B*, were found to be associated with litchi fruit abscission. Our results offer new knowledge and resources to study the function of plant ARF genes and their roles in the fruit abscission in litchi.

## MATERIALS AND METHODS

### Plant materials and treatments

The young fruitlet used for RNA-seq in our study were collected from 9-year-old litchi trees (*L. chinensis* Sonn. cv. "Feizixiao") in an orchard located at South China Agricultural University (Guangzhou, China). Three treatments have been performed 25 day after anthesis. The three treatments included ethephon (ETH), girdling plus defoliation (GPD), and dipping in 20 mg/L 2, 4-D for 1 min after girdling plus defoliation (GPDD). Details can refer to *Li et al. (2015)* and *Peng et al. (2013)*. Samples of "Feizixiao" (FZX) for qRT-PCR were obtained from the orchard of Guangzhou Fruit Research Institute (Guangdong, China). Tissues including fruit-bearing shoots (FBS), young leaves (YL), mature leaves (ML), female flower (FF), male flower (MF), undeterminated-sex flower (USF), and young fruit (YF) of 25 days after fertilization were collected from different directions of each tree. After separation, all tissues were quickly frozen in liquid nitrogen and stored at $-80\,^{\circ}$C before usage.

### Identification and phylogenetic analyses of LcARFs

Amino acid sequences of 25 and 23 ARFs from rice (*Wang et al., 2007*) and *Arabidopsis* (*Okushima et al., 2005*; *Wang et al., 2007*) were downloaded from UniProt (http://www.uniprot.org/) and 22 ARFs from tomato (*Zouine et al., 2014*) were downloaded from Sol Genomics Network (https://solgenomics.net/). These ARFs were used as bait to identify potential ARFs in litchi genome and obtained by BLAST analysis in TBtools (*Chen et al., 2018a*), with the cutoff *E*-value set at $1e^{-10}$. All sequences were then further validated by conserved domain search against the CDD (http://www.ncbi.nlm.nih.gov/cdd/) and PFAM (http://pfam.xfam.org/) databases. Based on the optimized alignment of amino acid sequences of LcARFs proteins, achieved by TrimAL 1.3 (http://phylemon2.bioinfo.cipf.es/index.html) after the initial sequence alignment in ClustalX 2.1, a maximum likelihood (ML) tree was constructed in MEGA 7.0 software with a bootstrap of 1,000 replicates. Intron and exon distribution patterns and genome structure were analyzed and visualized in TBtools (*Chen et al., 2018a*). The details of predicted protein sequences of LcARFs were shown in Datas S1 and S2.

### *LcARF* targets of miRNAs and RNA-seq analysis

Most of the *ARF*-targeting miRNA/tasiRNAs were identified in our previous study (*Ma et al., 2018*). To further validate the targeting relationship, target sites were verified by

psRNATarget (http://plantgrn.noble.org/psRNATarget) (*Dai & Zhao, 2011*) and miRBase (http://www.mirbase.org/).

RNA-seq analysis was carried on as described previously (*Li et al., 2013*). These RNA-seq data were normalized using the TPM method (*Wagner, Kin & Lynch, 2012*). Based on the sequences of the identified *LcARFs*, a BLASTN search was conducted to find their gene counterparts in previous data (*Li et al., 2013*). Finally, global gene expression profiles were visualized by heatmap via TBtools software (*Chen et al., 2018a*).

## Quantitative RT-PCR

Total RNA of the above seven tissue samples was extracted by column plant RNAout kit (Tiandz, Beijing, China) according to the manufacturer's instructions. About two μg total RNA was applied to synthesized first strand cDNA using reverse transcriptase AT311-03 (TransGen Biotech, Beijing, China). PCR primers (Table S1) of *LcARFs* and reference genes *GAPDH* and *EF* (*Zhong et al., 2011*) were designed by Primer Premier 5.0. qRT-PCR was performed according to the manufacturer's specifications of THUNDERBIRD qPCR MIX QPS-201 (Toyobo, Shanghai, China) on a LightCycler 480 (Roche, Rotkreuz, Switzerland). Each expression profile was independently verified in three biological replicates. Relative expression level of each gene was calculated by the $2^{-\Delta\Delta Ct}$ method (*Livak & Schmittgen, 2001*). Significance analysis was conducted in SPSS version 22.0 and visualized by SigmaPlot 12.5.

## RESULTS

### Identification and phylogenetic analysis of LcARF genes

To identify all ARF members in litchi, protein sequences of ARFs in rice, *Arabidopsis* and tomato were used as queries in BLASTP to search against a litchi annotated gene database. A total of 68 potential ARF genes were identified. After redundant result elimination and further conserved domain validation, 39 LcARFs were ultimately obtained. Among them, 23 protein sequences contained B3, ARF, and the Aux/IAA domains. Length of these litchi ARF proteins ranged from 260 (LcARF1B) to 1,200 (LcARF4A) and the relative molecular mass of them varied from 29,522.5 (LcARF1B) to 144,936.9 Da (LcARF16C), with PIs in the range of 5.28 (LcARF16C) to 8.64 (LcARF19A) (Table S2). According to the subcellular localization predictor (CELLO v.2.5; http://cello.life.nctu.edu.tw), most LcARFs were predicted to be located in the nucleus.

A phylogenetic tree was generated using the ML method based on alignment of litchi ARFs with their orthologs from rice, *Arabidopsis* and tomato (Fig. 1). All 109 ARFs fell into three broad groups: class I, II, and III, which contained 53, 34, and 22 members, respectively. Obviously, most ARFs of the four species were clustered together in the first two classes. Litchi ARFs were named according to their positions with orthologs from the other three species in the tree. Thus, the 39 LcARFs could also be assigned to three separate clusters as well. Class I contained LcARF1/2/3/4/9/18 (1A/B/C/D, 2A/B/C/D/E/F, 3A/B, 4A/B/C, 9A/B, 18A/B); Class II included LcARF5/6/7/8/19 (5A/B, 6A/B/C/D, 7A/B, 8A/B, 19A/B); Class III included LcARF10/16/17 (10A/B, 16A/B/C/D, 17A/B).
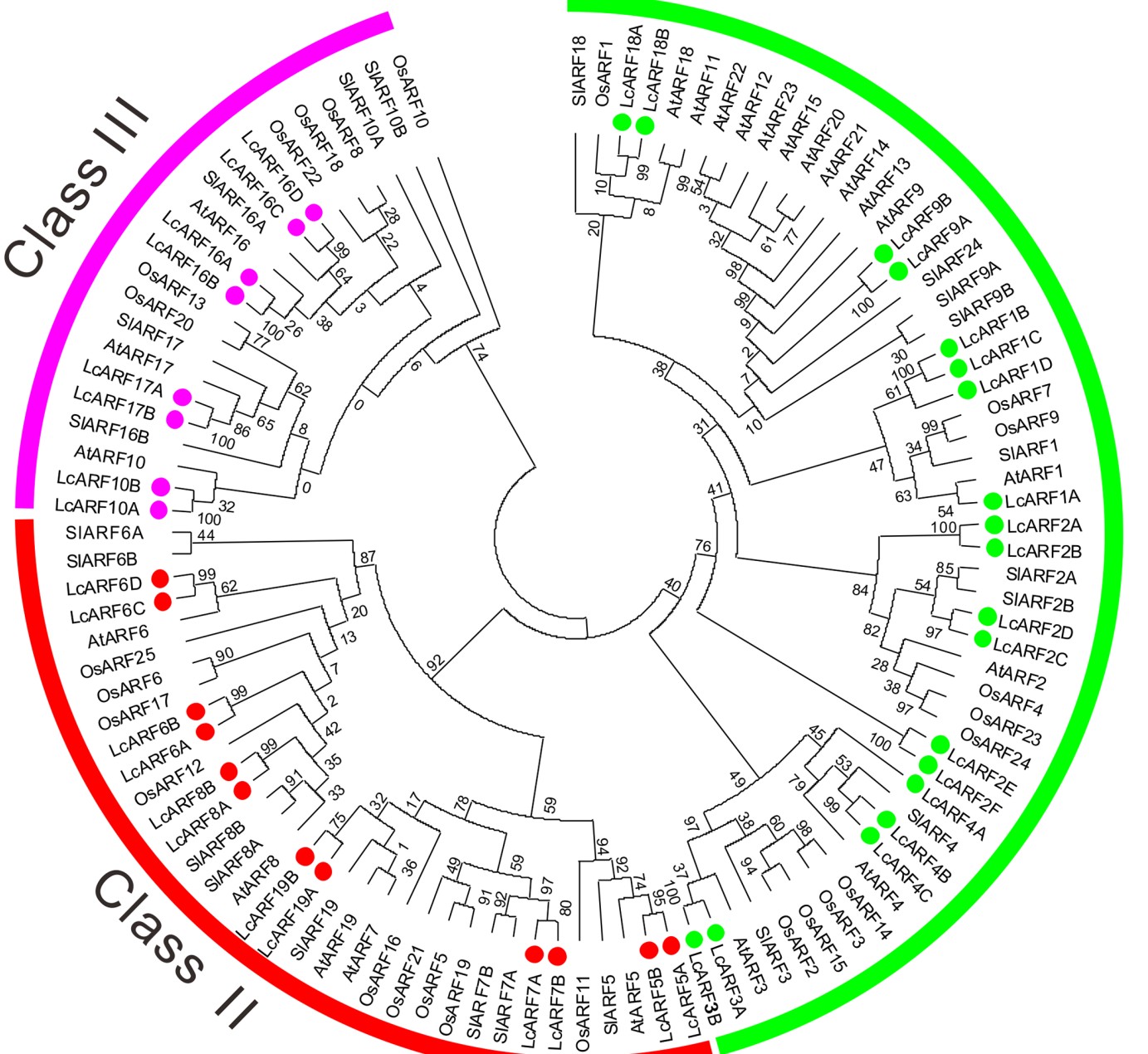

**Figure 1 Phylogenetic analysis of ARFs from litchi, rice, *Arabidopsis*, and tomato.** The phylogenetic tree was generated using the maximum likelihood method with the JTT matrix-based model (*Kumar, Stecher & Tamura, 2016*) and the bootstrap test was carried out with 1,000 bootstrap replicates. Numbers on the nodes indicate the credibility values of each clay. Three subgroups were shown as Class I, II, and III.

## Gene structure and conserved domains of litchi ARFs

To better understand the structure evolution of LcARF genes, their gene structure (intron/exon number and positions) and functional domains were analyzed. As shown in Fig. 2, 23 *LcARFs* harbored the typical ARF protein structure which composed of a highly conserved DBD in the N-terminal region with a plant specific B3-type subdomain,

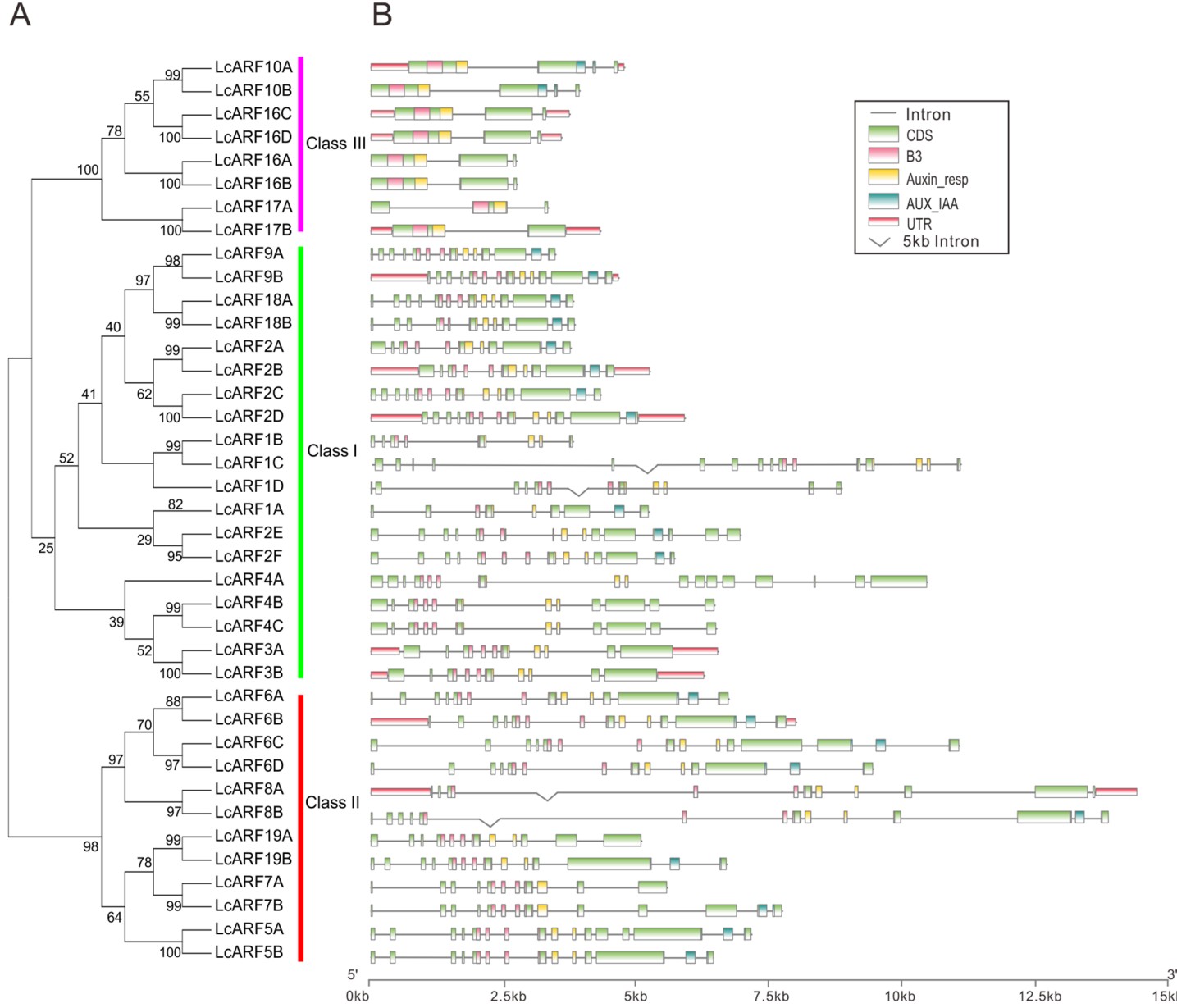

**Figure 2 Phylogenetic relationship, exon–intron structure, conserved domains analyses of LcARFs.** (A) Phylogenetic relationship among the litchi ARF proteins. The unrooted tree was generated using the maximum likelihood method by JTT matrix-based model. The reliability was assessed using 1,000 bootstrap replicates. Three clusters are labeled as Class I, Class II, and Class III. (B) Exon–intron structure and conserved domains of LcARFs. Information of exon, intron, and functional domain was obtained from model gene annotation and results of NCBI CDD search and visualized by TBtools. B3: B3 DNA-binding domain; Auxin-resp: ARF domain; AUX_IAA: C-terminal dimerization domain. Lengths of exons and introns and domains of each LcARF protein were exhibited proportionally.

an Auxin-resp subdomain, and an AUX_IAA dimerization subdomains. The remaining 16 *LcARFs* contained only B3 and Auxin_resp subdomains. Gene structure analysis revealed that the exon number of class III (2–4 exons) was significantly less than the other two groups (8–17 exons for class II and 10–16 exons for class I). These results provided additional evidence confirming the phylogenic relationships among LcARFs.

## Analyses of miRNA targeting *LcARFs*

A total of 22 out of the 39 LcARF genes were found to be the targets of miRNA (Fig. 3). All group members of class III (*LcARF10A/B*, *16A/B/C/D*, and *17A/B*) were found to be targeted by Lc-miR160. *LcARF6A/B/C/D* and *8A/B* were members of class II and all of them were collectively targeted by Lc-miR167. Two kinds of miRNA targeting patterns were observed in the class I. One group comprising *LcARF2E/2F/4B/4C/3A/3B* were targeted by tasiARFs. In litchi, *LcTAS3* are divided into two subgroups, long TAS3 genes (*LcTAS3_1* and *LcTAS3_2*) and short TAS3 genes (*LcTAS3_3* and *LcTAS3_4*), which trigger to produce one or two tasiARFs when cleaved by miR390 (*Ma et al., 2018*). Additionally, *LcARF2E/2F* incorporated one target site of tasiARF, while *LcARF3A/3B/ 4B/4C* contained two, which is in accordance with our previous study (*Xia, Xu & Meyers, 2017*). Interestingly, in the other group, a novel miRNA-*ARF* pathway was discovered, in which *LcARF9A/B* was targeted by the Lc-miRN43.

## Expression of LcARF genes in different organs and tissues

A large body of evidence support the importance of ARF genes in plant growth and development (*Ellis et al., 2005*; *Guilfoyle & Hagen, 2007*; *Lim et al., 2010*; *Li et al., 2016*). To explore how LcARF genes function in the development of litchi, we examined their expression levels in various litchi organs/tissues by qRT-PCR. Seven organs/tissues samples, composing FBS, MF, FF, USF, ML, YL, and YFs in "FZX" were collected and tested. As shown in Fig. 4A, expression of all *ARFs* was detected at least in several tissues. Generally, most *LcARFs* show low expression in YL and YF but high in USF. *LcARFs* from different clades showed various expression patterns. *LcARFs* from class I and III were detected to be higher expressed in ML while those from class II were lowly accumulated, suggesting that these *LcARFs* in different classes may perform different functions in ML development (Fig. 4A). Additionally, as is shown in Fig. 4B, 10 *LcARFs* (*LcARF2B*, *3A*, *4A/B*, *5A/B*, *6D*, *9A/B*, *18B*) were significantly higher expressed in FF than MF, implying their potential function in ovule and ovary might be derived from FFs. Remarkably, three *LcARFs* (*LcARF8A*, *LcARF10A/B*) were of significantly higher expression in MF than FF, indicated that they might play roles in MF formation. Additionally, a table for LcARFs and their potential function was concluded in Table S3 according to an early study which may be useful for further functional validation.

## Expression profiles of LcARF genes in response to ETH, GPD, and GPDD treatment

Auxin is a critical signal in the abscission of fruits in plants (*Blanusa et al., 2005*; *Meir et al., 2010*; *Xie et al., 2013*). To explore the role of *LcARFs* in fruitlet abscission in litchi, RNA-seq was carried on and transcription levels of LcARF genes in the abscission zone were investigated under three treatments (ETH, GPD, GPDD). Both the ETH and GPD treatments promote fruitlet abscission while GPDD delays the process (*Peng et al., 2013*; *Li et al., 2015*). As shown in Fig. 5, the transcript expression of most *LcARFs* were decreased after treated by ETH and GPD, but an upsurge was present after GPDD treatment, which was corresponding to the process of promoting or inhibiting of fruitlet

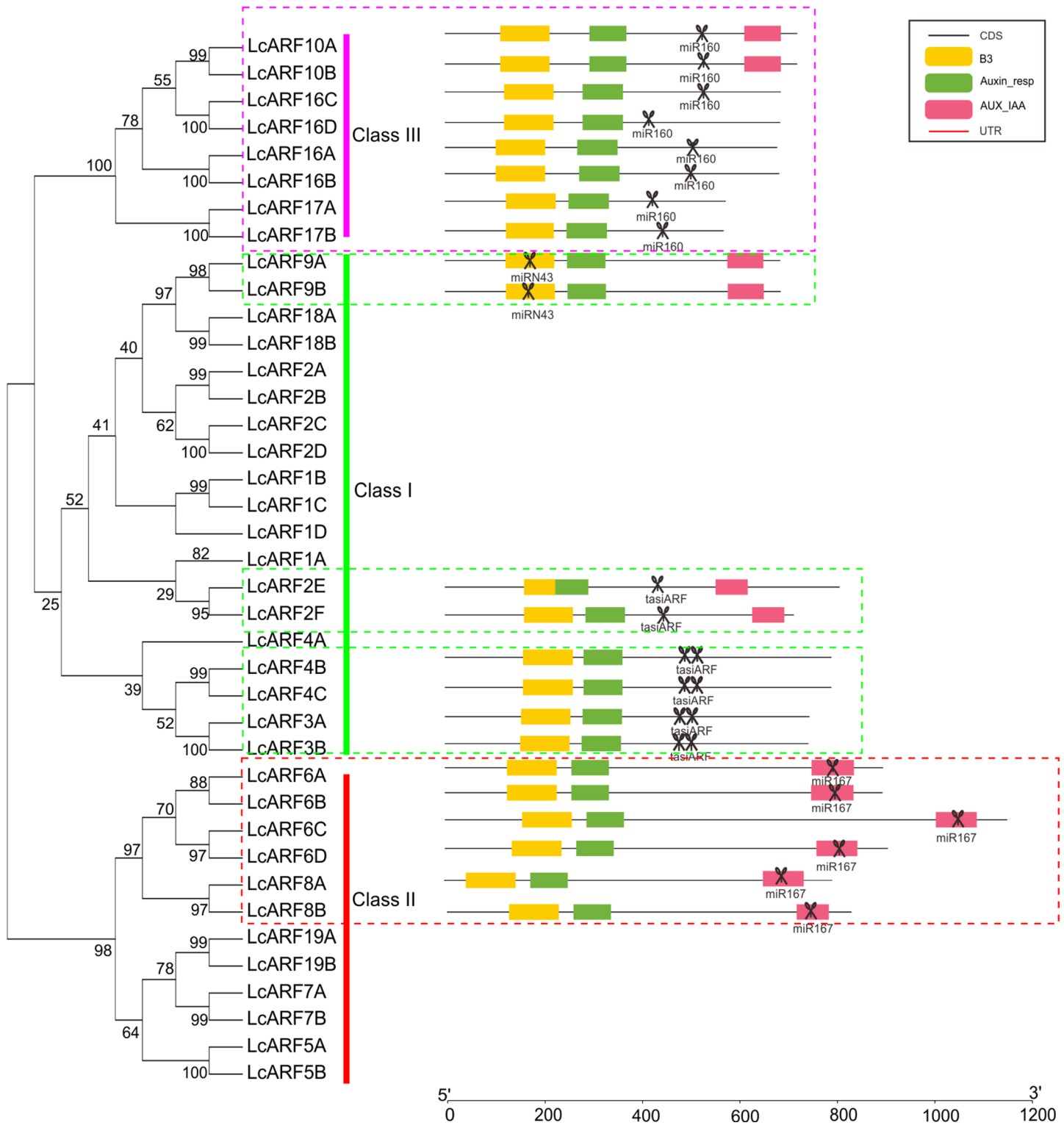

**Figure 3 LcARFs targeted by some LcmiRNAs.** miR160 targets *LcARF10/16/17*, miR167 targets *LcARF6/8,* and miRN43 targets *LcARF9*. Additionally, the *LcTAS3* was targeted by miR390 and then triggered the production of tasiARF which target *LcARF2E/2F/4B/4C/3A/3B*.

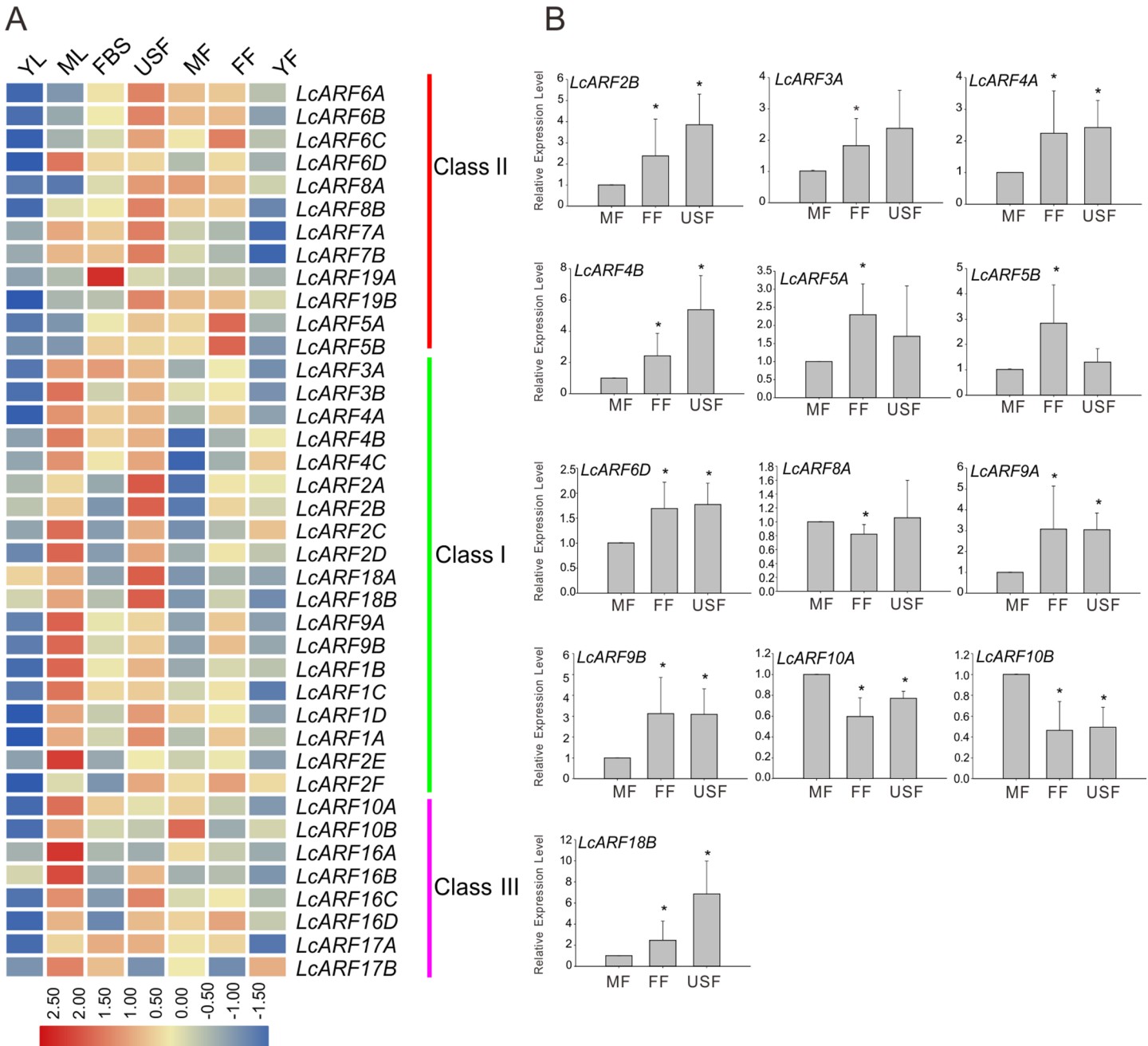

**Figure 4 Expression profiles of LcARF genes in various tissues of "FZX" by qRT-PCR.** (A) Expression of all *LcARFs* in different tissues. The heatmap was generated based on the relative expression values of 39 *LcARFs* obtained by qRT-PCR in seven different tissues and organs. Red and blue were represented relatively high and lower expression (log$_2$ ratio), respectively. Every sample has three biological replicates. YF, young fruit (25 days after fertilization); FF, female flower; MF, male flower; USF, undetermined-sex flowers; ML, mature leaves; YL, young leaves; FBS, fruit-bearing shoots. (B) Relative abundance of *LcARFs* significantly expressed in FF. Data from three independent biological replicates are shown with standard deviation (SD). Asterisks on the top of bars indicate significant differences as determined by Student's *t*-test (*$P$, 0.05).

shedding, respectively. It seemed that there was a negative correlation between *LcARFs* and the fruit abscission and among these *LcARFs*, those from two groups (group 2 and group 3) were particularly representative. Moreover, *LcARFs* from group 3 were more sensitive to
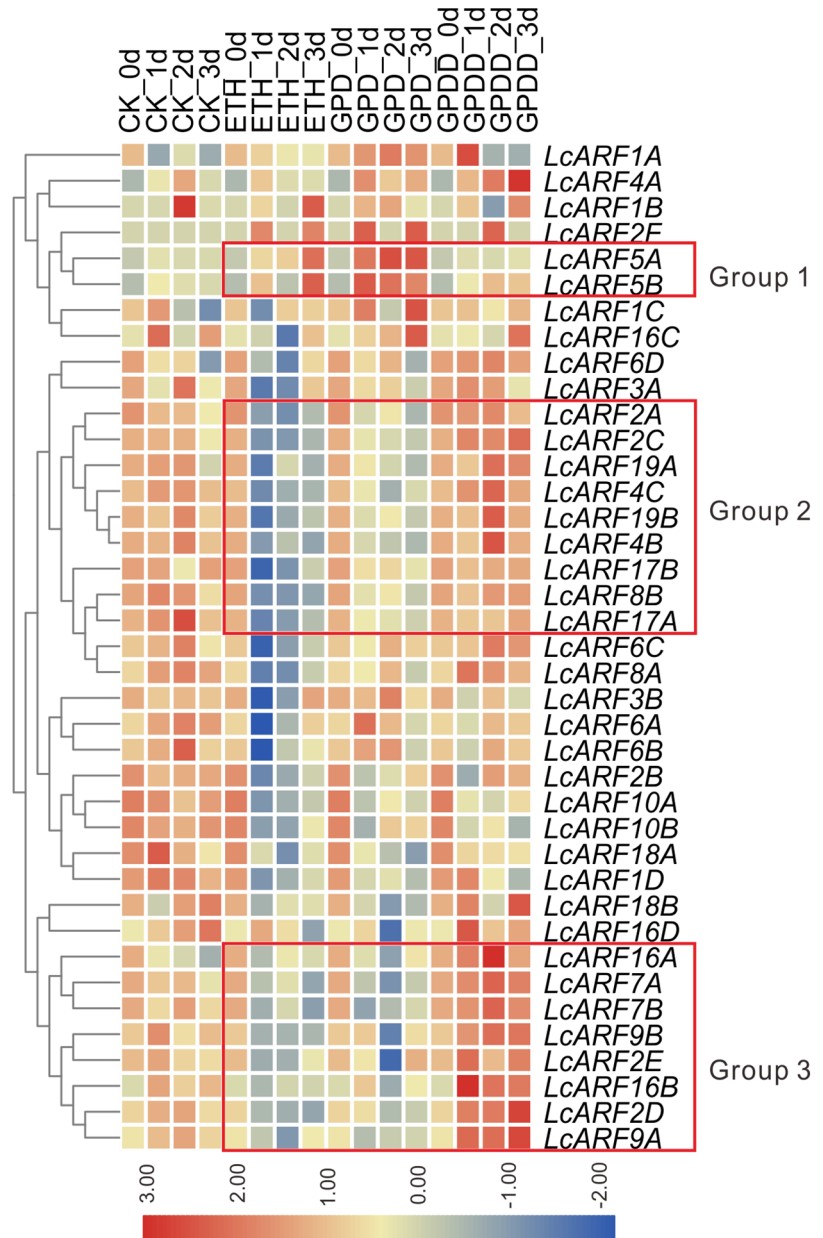

**Figure 5 Expression profiles of LcARF genes in response to ETH, GPD, and GPDD treatments.**
Fruit-bearing shoots of "FZX" litchi were obtained at 25 days after anthesis and then carried on ETH, GPD, and GPDD treatment from 0, 1, 2, and 3 days, respectively. CK, control; ETH, treated by ethylene; GPD, girdling plus defoliation; GPDD, dipping in 2, 4-D after GPD treatment. The heatmap was created based on the TPM values of *LcARFs* from the transcriptome data. In the heatmap, red and blue were represented higher and lower expression (log$_2$ ratio), respectively. Heatmap and hierarchical clustering were performed by average linage (default) method.

GPDD treatment along with stronger expression than those from group 2. Thus, we could deduce that *LcARFs* from group 3, including *ARF2D/2E, 7A/7B, 9A/9B, 16A/16B*, played major roles on the litchi fruitlet abscission. In contrast, a few *LcARFs*, such as group 1 including *LcARF5A/5B*, showed an opposite expression pattern, suggesting that they might function to accelerate the process of abscission. There were as well

some LcARFs with no significant expression change and seemed to be unrelated to fruitlet abscission.

## DISCUSSION

Litchi is an important tropic fruit tree and massive fruit abscission before harvest usually leads to low and even no production. Auxin is proposed to be one of the endogenous hormones playing significant roles in the regulation of fruit abscission in litchi (*Yuan, 1988*; *Stern & Gazit, 2000*). In this study, 39 LcARF genes were identified, which was found more than in the other model plants, such as *Arabidopsis* (23) (*Okushima et al., 2005*; *Wang et al., 2007*), rice (25) (*Wang et al., 2007*), and tomato (22) (*Zouine et al., 2014*), implying extensive duplication and diversification of the ARF gene family in litchi. Analysis of conserved motifs revealed that all LcARFs had a typical DBD domain required for efficient binding to AuxRE and a Auxin_resp (Fig. 2) (*Hagen & Guilfoyle, 2002*; *Ha et al., 2013*). However, only 23 of 39 LcARFs contain the AUX_IAA domain, which can mediate the dimerization of ARFs or ARF and Aux/IAA protein (*Guilfoyle & Hagen, 2007*). Lack of the AUX_IAA domain for dimerization makes it interesting to address questions like how these ARFs function and whether they need dimerization with other proteins. In plants, ARFs can function as transcription activators (ADs) or repressors (RDs), according to the amino acid composition of Auxin_resp domain (*Guilfoyle & Hagen, 2007*). In *A. thaliana*, ARF ADs and RDs were proposed to contain biased amino acid sequences, which ARF ADs were enriched in glutamine (Q), while RDs were enriched in serine (S), serine and proline (SP), and serine glycine (SG) (*Guilfoyle & Hagen, 2001*; *Tiwari, Hagen & Guilfoyle, 2003*). Intriguingly, no ARFs in litchi were enriched in Q, but with SPL and SP/SG enrichment; therefore none of LcARF proteins seems to be an activator (reviewed in *Guilfoyle & Hagen (2007)*). Further experiments are needed to verify this observation.

Much evidence demonstrates that miRNAs play essential roles in post-transcriptional gene regulation in plants (*Jones-Rhoades, Bartel & Bartel, 2006*; *Li & Zhang, 2016*). It has been found that several ARF genes are regulated by a few miRNAs. In our work, 22 out of 39 LcARF genes were found to be targets of miRNAs (Fig. 3) and *LcARFs* from different classes were displayed in different miRNA targeting patterns. Members of class III (*LcARF10A/B*, *16A/B/C/D*, and *17A/B*) were found to be targeted by Lc-miR160, which might affect flower development of litchi, as down-regulation of *ARF10/16/17* by miRNA160 is reported to regulate floral organ abscission in tomato (*Damodharan, Zhao & Arazi, 2016*). Consistent with *Arabidopsis*, *LcARF6/8* were collectively targeted by Lc-miR167. Interestingly, even though it has been reported that *LcARF8B* is targeted by Lc-miR167 (*Ma et al., 2018*), *LcARF8A* was a novel target by Lc-miR167. Notably, in another group, a novel miRNA-*ARF* pathway was discovered in litchi, in which *LcARF9A/B* was targeted by Lc-miRN43. In fact, miRN43 is an innovative miRNA as well, for it is unable to find any ortholog in the database of miRase, which may provide a new interaction for us to study for the specific functions within litchi.

The expression profiles can help us screen for candidate *LcARF* genes with potentially distinct functions. Most *LcARFs* were higher accumulated in ML than YL including

*LcARF2,* and its Arabidopsis homolog, AtARF2, which has been reported to regulate leaf senescence with high expression in ML (*Ellis et al., 2005*). Thus, we can deduce that LcARF2 is likely to be involved in leaf senescence in litchi. It has been reported that Arabidopsis ARF genes regulate flower formation, especially gametophyte development, which are critical for sexual reproduction. AtARF2–AtARF4 and AtARF5 play significant roles in regulating both female and male gametophyte development in Arabidopsis (*Liu et al., 2017*) and ARF6 and ARF8 regulate both stamen and gynoecium maturation (*Nagpal, 2005*; *Ru et al., 2006*; *Wu, Tian & Reed, 2006*; *Liu et al., 2014b*). Thus their litchi homologs LcARF2B, 3A, 4A/B, 5A/B, 6D, and 8A may be important for female and male gametophyte development in litchi, along with significantly higher accumulation in female or MF. Moreover, *LcARF4, 5* and *9* were prominently higher expressed in FF than MF, implying their potential function in ovule and ovary development.

Auxin response factor genes have been reported to be involved in plant organ abscission. Overexpression of *SlARF2* in tomato results in flower organ senescence (*Ren et al., 2017*) and SlARF1, 2, 7, 11, and 19 showed overlapping functions in tomato abscission (*Guan et al., 2014*). Similar roles of ARFs in abscission were observed in *Arabidopsis* (*Ellis et al., 2005*). Our previous studies show that the treatment of GPD in litchi could reduce the transcript level of auxin response factor (*LcARF1*) mRNA, along with the increase of fruitlet abscission (*Kuang et al., 2012*). Here in our RNA-seq survey of ARF gene expression, we found that most of the ARF genes show an opposite correlation with the fruit abscission, that is, ARF genes were down-regulated by abscission induced treatments (ETH and GPD), but up-regulated by GPDD, in which the abscission of GPD was inhibited by the addition of 2, 4-D. This suggested that the majority of LcARF genes were negatively involved in litchi fruit abscission, especially *LcARFs* from group 3 (*ARF2D/2E, 7A/7B, 9A/9B, 16A/16B*), with more prominent expression after GPDD treatment. By contrast, a few *LcARFs*, such *LcARF5A/5B*, showed positive correlation with fruit abscission, indicating that they might serve as contributing factors to fruit shedding.

## CONCLUSION

In this study, a total of 39 ARF genes were identified from the litchi genome. Comprehensive analyses, including phylogenetic relationship, exon–intron structure, conserved domain, and potential targets for small RNAs, revealed that the ARF gene family was expanded in litchi with species-specific features. A novel miRNA-*ARF* (miRN43-*ARF9*) regulatory pathway was discovered, which is likely specific in litchi. Expression profiles in various organs and under different abscission-related treatments (ETH, GPD, and GPDD) uncovered the expression diversity of these ARF genes in litchi. Some ARF genes, including *ARF2D/2E, 7A/7B, 9A/9B, 16A/16B,* and *5A/5B*, likely play predominant roles in the process of litchi fruit abscission. These findings provide new knowledge and resources for further functional characterization of ARF genes in litchi.

## Funding

This work was supported by the National Natural Science Foundation of China (No. 31872063, 31471859), the Innovation Team Project of the Department of Education of Guangdong Province (No. 2016KCXTD 011), the Guangzhou Science and Technology Key Project (No. 201804020063), the Outstanding Talent Program of the Ministry of Agriculture and the China Agricultural Research System (No. 201804020063). The funders had no role in study design, data collection and analysis, decision to publish, or preparation of the manuscript.

## Grant Disclosures

The following grant information was disclosed by the authors:
National Natural Science Foundation of China: 31872063, 31471859.
Innovation Team Project of the Department of Education of Guangdong Province: 2016KCXTD 011.
Guangzhou Science and Technology Key Project: 201804020063.
Outstanding Talent Program of the Ministry of Agriculture and the China Agricultural Research System: 201804020063.

## Competing Interests

The authors declare that they have no competing interests.

## Author Contributions

- Yanqing Zhang performed the experiments, analyzed the data, prepared figures and/or tables, authored or reviewed drafts of the paper, approved the final draft.
- Zaohai Zeng prepared figures and/or tables, authored or reviewed drafts of the paper, approved the final draft.
- Chengjie Chen analyzed the data, contributed reagents/materials/analysis tools, approved the final draft.
- Caiqin Li conceived and designed the experiments, performed the experiments, approved the final draft.
- Rui Xia conceived and designed the experiments, authored or reviewed drafts of the paper, approved the final draft.
- Jianguo Li conceived and designed the experiments, authored or reviewed drafts of the paper, approved the final draft.

## Data Availability

All raw data used for the transcriptome analysis and database construction are available at Sequence Read Archive (SRA) under Project ID PRJNA509724.

## Supplemental Information

Supplemental information for this article can be found online at http://dx.doi.org/10.7717/peerj.6677#supplemental-information.

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
