# Peer review of "Genome-wide characterization of the auxin response factor (ARF) gene family of litchi (Litchi chinensis Sonn.): phylogenetic analysis, miRNA regulation and expression changes during fruit abscission"

_PeerJ, doi:10.7717/peerj.6677_

## Round 0.1 · original submission · Minor Revisions

Please revised the manuscript according to the comments from the reviewers.

Reviewer 1 ·

Basic reporting

My comments are included in "General comments for the author".

Experimental design

My comments are included in "General comments for the author".

Validity of the findings

My comments are included in "General comments for the author".

Additional comments

Zhang et al. reported comprehensive identification of auxin response factor (ARF) genes from lirchi, a commercially important fruit tree, especially in China. The litchi ARF proteins were named according to the phylogenetic analyses with ARFs from other model plants, Arabidopsis, rice and tomato. The introns and exons structures of the genes were compared among LcARFs, and conserved domains of ARF were identified. The authors also found target sites of miRNAs in the LcARF genes and discovered a novel miRNA pathway to regulate LcARF9s expression. The authors analyzed expression of all LcARFs in various organs/tissues by qRT-PCR and also analyzed effects of abscission-inducing treatments by RNA-seq. Throughout the manuscript, the experiments were well done and the results were satisfactorily and concisely described. However, the profiling of RNA seq data should be re-analyzed with a normalization method other than RPKM because comparison among samples with RPKM has been demonstrated to be inappropriate (Wagner GP, Kin K, Lynch VJ. Measurement of mRNA abundance using RNA-seq data: RPKM measure is inconsistent among samples. Theory Biosci. 2012, 131(4):281-5.).

Other points
1. In line 162, “Class I” should be collected as “Class II” according to Fig. 1.
2. In line 195, the authors described that “all LcARF genes were expressed in all organs/tissues studied”, although in some tissues, the expression levels are very low based on the results in Fig. 4. Generally, by qRT-PCR, it is difficult to distinguish very low accumulation of mRNAs from contaminated genomic DNAs or non-specific amplification. In the context of the manuscript, it is unlikely necessary to claim the significant expression in all tissues. The authors can mention that expression of all ARFs was detected at least in several tissues, which still means that all ARF genes found in this study are likely functional but not pseudo genes.
3. In line 197-199, the authors described that the class I and class III ARFs were expressed at higher levels in “ML”. The authors have defined that ML is “mature leaves”, thus these genes may function in mature leaves but not in “male flower development”, which is described in line 199.
4. In line 242, the authors described none of the LcARFs seems to be activator. As the authors mentioned, there may be no activators with glutamine rich AD, but the data could not exclude the possibility that some ARFs may have other types of ADs.
5. Please indicate SE or SD for the error bars in Fig. 4B.
6. In the graph for LcARF9B in Fig. 4B, the numbers of Y-axis are not located at equal interval. I don’t know the journal rule, but I have never seen the numbers in axes of graphs showing as “.5” and like that.

Reviewer 2 ·

Basic reporting

'no comment

Experimental design

'no comment

Validity of the findings

Please do some RACE to confim some miRNA target ARFs

Additional comments

The article is in good writing. But "phylogenetic analysis" and "evolution" are much different. There are some TasiRNA in “miRNA regulation” . Please modify your title of manuscript.

·

Basic reporting

no commen

Experimental design

no comment

Validity of the findings

no comment

Additional comments

This manuscript has elaborated transcriptional analysis based on qRT-PCR of LcARF gene family. It predicts important roles of these LcARF genes in various tissues/organs/zones and hormone-treated conditions. The authors conducted an interesting investigation on a topic that is of interest to several plant biologists and hence shall invite a broad range of audience. The quality of work and illustrations are excellent. I recommend this for publication after the authors carry out some minor revisions raised by me. The revisions will improve the readability of the paper.
1. Authors should provide the resource of Litchi genomic database. The detail of predicted protein sequences of LcARFs should also provide.
Authors also should indicate some report of characterization of ARF family in important crops such as rice, maize, soybean in introduction paragraph.
2. Nomenclature for LcARFs should use no.1--> 39. The chromosome location of LcARFs should include in Table S1.
Gene names should be corrected in Italic in Figures 4,5, table S1 and main text of the manuscript.
3. Authors should discuss more of the evolution as well as the tissue specific of the LcARFs to identify some potential genes for further analysis.
4. Authors should include a conclusive table for the important genes and their function, observed in the transcriptome study in tissue/zone specific and hormone condition, that can be used for further functional validation.
5. Author should be correct the English, format errors and missing of citation. For example, length of proteins (aa), molecular mass (Da) L151-154, italic format for the gene name.

---

## Round 0.2 · accepted · Accept

The reviewers satisfied the revised version of manuscript and recommended acceptance of the manuscript. I agree with the recommendation made by the reviewers.

# EDITS LINE 26: / economical / economically /
LINE 30: / Number / The number /
LINE 32: / , III / , or III /
LINE 33: / Totally, / In total, /
LINE 38: / might positively involve / might be positively involved /
LINE 67: / defectives / defects /
LINE 85: / involved / is involved / [Either sentence is incomplete or “is” is needed.]
LINE 89: / trees / tree | tree resource /
LINE 93: / other in / other ARFs in /
LINE 120: / Blast / BLAST /
LINE 121: / E-value / cutoff E-value set at /
LINE 121: / by conserved / by a conserved /
LINE 136: / was normalized using / were normalized using the /
LINE 138: / a global / global /
LINE 142: / of above / of the above /
LINE 149: / level / levels /
LINE 150: / vision / version /
LINE 151: / visulized / visualized /
LINE 156: / against litchi / against a litchi /
LINE 158: / obtained ultimately / ultimately obtained / [?]
LINE 163: / prediction, most / , most /
LINE 194: / two or one / one or two / [Why in the inverse order?]
LINE 214: / derived from / may be derived from /
LINE 217: / concluced / concluded /
LINE 217: / to early study / to an early study /
LINE 226: / Fig. 5, / Fig. 5 demonstrate, /
LINE 227: / upsurge / an upsurge was present/
LINE 234: / opposite / an opposite /
LINE 238: / Discussions / Discussion /
LINE 239: / tropic fruit trees / tropical fruit tree /
LINE 242: / more than / found more than in /
LINE 247: / contain / contained the /
LINE 255: / serine glycine / and serine glycine /
LINE 256: / then / therefore /
LINE 257: / 2007)). / 2007). /
LINE 264: / different miRNA targeting pattern. / in different miRNA targeting patterns. /
LINE 268: / there has been / where it has been /
LINE 272: / new idea / new interaction /
LINE 273: / the specific functions of / for the specific functions within /
LINE 277: / AtARF2 has been / , AtARF2, which has been /
LINE 279: / to involve / to be involved /
LINE 287: / prominent / prominently /
LINE 296: / ARF genes expression / ARF gene expression /
LINE 297: / opposite correlation / an opposite correlation /
LINE 299: / 2, 4-D, suggesting / 2,4-D. This suggested / [Break-up run-on sentence.]
LINE 300: / are negatively / were negatively /
LINE 302: / show / showed /
LINE 305: / Conclusions / Conclusion /

These were merely suggestions - a final editing and re-read should provide a better delivery to the PeerJ readers.

Reviewer 1 ·

Basic reporting

no comment.

Experimental design

no comment.

Validity of the findings

no comment.

Additional comments

The authors have positively addressed my concerns and the revised manuscript has been well improved. I have no further comment.

Reviewer 2 ·

Basic reporting

no comment

Experimental design

no comment

Validity of the findings

no comment

Additional comments

all concerns have been addressed